# A Novel in Duck Myoblasts: The Transcription Factor Retinoid X Receptor Alpha (RXRA) Inhibits Lipid Accumulation by Promoting CD36 Expression

**DOI:** 10.3390/ijms24021180

**Published:** 2023-01-07

**Authors:** Ziyi Pan, Xingyong Chen, Dongsheng Wu, Xuewen Li, Weifeng Gao, Guoyu Li, Guoqing Du, Cheng Zhang, Sihua Jin, Zhaoyu Geng

**Affiliations:** College of Animal Science and Technology, Anhui Agricultural University, Hefei 230036, China

**Keywords:** transcription factor, RXRA, CD36, lipid metabolism, myoblasts

## Abstract

Retinoid X receptor alpha (RXRA) is a well-characterized factor that regulates lipid metabolism; however, the regulatory mechanism in muscle cells of poultry is still unknown. The overexpression and the knockdown of RXRA in myoblasts (CS2 cells), RT-PCR, and western blotting were used to detect the expression levels of genes and proteins related to PPAR-signaling pathways. Intracellular triglycerides (TGs), cholesterol (CHOL), and nonesterified free fatty acids (NEFAs) were detected by the Elisa kit. Fat droplets were stained with Oil Red O. The double-fluorescein reporter gene and chromatin immunoprecipitation (CHIP) were used to verify the relationship between RXRA and candidate target genes. The RXRA gene was highly expressed in duck breast muscle, and its mRNA and its protein were reduced during the differentiation of CS2 cells. The CS2 cells, with the overexpression of RXRA, showed reduced content in TGs, CHOL, NEFAs, and lipid droplets and upregulated the mRNA expression of CD36, ACSL1, and PPARG genes and the protein expression of CD36 and PPARG. The knockdown of RXRA expression in CS2 cells enhanced the content of TGs, CHOL, NEFAs, and lipid droplets and downregulated the mRNA and protein expression of CD36, ACLS1, ELOVL6, and PPARG. The overexpression of the RXRA gene, the activity of the double-luciferase reporter gene of the wild-type CD36 promoter was higher than that of the mutant type. RXRA bound to −860/−852 nt, −688/−680 nt, and −165/−157 nt at the promoter region of CD36. Moreover, the overexpression of CD36 in CS2 cells could suppress the content of TGs, CHOL, NEFAs, and lipid droplets, while the knockdown expression of CD36 increased the content of TGs, CHOL, NEFAs, and lipid droplets. In this study, the transcription factor, RXRA, inhibited the accumulation of TGs, CHOL, NEFAs, and fat droplets in CS2 cells by promoting CD36 expression.

## 1. Introduction

In the past 20 years, the world’s demand for poultry meat protein has continued to increase, and the need for duck meat has also increased. Asia produced nearly 3 million tons of duck meat in 2017, accounting for 75% of the world’s production [1]. Human diet habits prefer to choose high-quality duck meat. Intramuscular fat (IMF) content plays a key role in determining duck meat quality because it affects duck meat flavor, juiciness, tenderness, and muscle color [2]. Duck IMF is composed primarily of triglycerides (TGs), cholesterol (CHOL), and fatty acids (FAs) [3,4]. Fatty acids are the raw materials for the synthesis of TGs and CHOL esters [5] (Pikul et al., 1984). FAs have various functions in the body. Increasing the contents of unsaturated FAs in the human diet can reduce the risk of cardiovascular disease and diabetes [6,7]. FAs also have a large effect on target tissue; NEFA-induced beta cell insulin secretion has been widely recognized [8]. However, too many FAs are key factors leading to obesity and related diseases. The excessive accumulation of FAs in the body is known to reduce glucose utilization and decrease glucose uptake [9]. The accumulation of FAs causes insulin resistance and dysfunction within myoblasts [10]. Therefore, controlling the FA metabolism of meat ducks has become an important research topic in myoblasts and FA accumulation and human health diet in recent years.

Retinoid X receptor A (RXRA), a member of the nuclear hormone receptor superfamily, is a transcription factor that regulates the expression of genes involved in cellular lipid metabolism [11]. RXR is able to bind to dietary fatty acids to participate in the tissue-specific expression of downstream target genes involved in lipid homeostasis [12]. The RXR transcription complex plays a key role in FA removal and glucose homeostasis [13]. It has been reported that RXR agonists are insulin sensitizers and can reduce blood sugar, hypertriglyceridemia, and hyperinsulinemia [14]. RXRA is a transcription factor that controls lipid metabolism by activating target gene transcription through PPRE response elements [15]. For example, RXR and PPARG form heterodimers to promote C/EPBA transcription and combine with PPARG to stimulate brown adipocyte differentiation. RXR genes also have their own unique and complex functions [16]. The RXRG is involved in common diseases of dyslipidemia and obesity in mice [17]. However, the effect and mechanism of RXRA on lipid metabolism in duck myoblasts have not been extensively studied.

Lipid metabolism is one of the most important factors impacting duck quality and is a major factor affecting consumers’ assessments of meat quality. However, the mechanism of the RXRA-regulating lipid metabolism in duck myoblasts remains unclear. Therefore, the duck myoblasts were cultured in vitro to investigate the effects of RXRA and its target gene on lipid metabolism. The target relationship between RXRA and its target genes was verified by the double-luciferase reporter gene and CHIP experiment. The transcription factor, RXRA, regulates the lipid metabolism of duck myoblasts by regulating target genes, providing ideas for the study of lipid metabolism in meat ducks.

## 2. Results

### 2.1. Expression Pattern of RXRA Gene in Duck Myoblasts

To evaluate the gene expression patterns of RXRA in various tissues of duck, RT-PCR was performed in 42 tissues. RXRA showed the highest expression in the breast muscle (Figure 1A). PAX7 is expressed in the nucleus as a marker protein of myoblasts. To investigate the role of RXRA in muscle, the location of RXRA in myoblasts (CS2 cells) was first determined, and the immunofluorescence analysis showed that RXRA was located in the nucleus of CS2 cells (Figure 1B). PAX7-positive, MYOD-positive, and RXRA-positive cells account for more than 95% of living cells, and the ratio of RXRA-positive cells to PAX7-positive and MYOD-positive cells is higher than 95% (Figure 1B). Next, the expression profiles of the RXRA gene during the differentiation of CS2 cells were characterized. The mRNA and protein expression levels of RXRA gene were decreased on the 6th day of differentiation (Figure 1C–E). The contents of TG, CHOL, and nonesterified fatty acid (NEFA) in CS2 cells increased on the 6th day of differentiation (Figure 1F–H). The RXRA gene plays an important role in the lipid metabolism of myoblasts.

### 2.2. The RXRA Gene Inhibits the Accumulation of TGs, CHOL, and NEFAs in Myoblasts

In order to explore the function of transcription factor RXRA in CS2 cells, RXRA mRNA and protein were effectively overexpressed and knocked down. Transfection results observed under fluorescence microscopy are shown in Appendix A. The RT-PCR results showed that, compared with cells transfected with empty vector, RXRA was effectively overexpressed and knocked down after oeRXRA and shRXRA transfection, separately (Figure 2A). Then, western blot analysis was performed to verify the level of RXRA protein expression. RXRA overexpression in CS2 cells transfected with oeRXRA was confirmed by western blot analysis (Figure 2B,C). Moreover, RXRA expression was effectively knocked down after shRXRA transfection (Figure 2B,C).

This paper investigates the effect of the RXRA gene on the lipid content of CS2 cells. Compared with that in the normal control vector groups, the TG content in the oeRXRA group was reduced, and the TG content in the shRXRA group was increased (Figure 2D). The CHOL content in the oeRXRA group was lower than that in the oeNC group, while the CHOL content in the shRXRA group was higher (Figure 2E). In addition, the changes in the NEFA content in CS2 cells were detected, and the results showed that the NEFA content in the oeRXRA group was decreased compared with that in the oeNC group (Figure 2F). In contrast, the NEFA content in the shRXRA group was elevated compared with that in the shNC group (Figure 2F). Lipid droplets quantification through Oil Red O staining showed that the number of fat droplets in the oeRXRA group shrank compared with the oeNC group. However, the number of fat droplets was enlarged after the knockdown of RXRA (Figure 2G,H).

### 2.3. RXRA Promotes Gene Expression in PPAR-Signaling Pathway

In this study, in order to explore the mechanism of RXRA’s affecting lipid metabolism through PPAR-signaling pathway, the expression of related genes and proteins were detected. As shown in Figure 3A, the relative expression levels of the CD36, ACSL1, PPARG, and FAS genes in the oeRXRA group were promoted, and there was no change in the expression levels of the FABPA, ELOVL6, and FASN genes between oeRXRA with the oeNC group. The CD36, ACSL1, ELOVL6, PPARG, and Cpt2 gene expression levels were inhibited by shRXRA, and there was no significant change in the FABPA, FAS, and FASN gene expression levels between the shRXRA group and the shNC group (Figure 3A). The results of the western blot analysis were consistent with the results of the gene mRNA expression analysis (Figure 3B,C). Interestingly, RXRA may have some connection with CD36.

### 2.4. RXRA Promotes CD36 Expression by Binding to the CD36 Promoter

According to above results, there were five RXRA-binding sites in the 2000 bp promoter region of target gene CD36, which were predicted by PROMO software to be expressed downstream of RXRA (Figure 3D). Double-luciferase activity and CHIP were detected, which verifies that RXRA binds to the promoter region of CD36. The results of co-transfection showed that oeRXRA increased the wt-CD36 promoter luciferase activity compared with the transfection of mu-CD36, GLP3-Basic, or oeNC. However, the transfection of oeNC exerted no significant effect on wt-CD36, mu-CD36, or GLP3-Basic luciferase activities (Figure 3E). The results showed that RXRA could promote the expression of CD36 by binding to five sites in the CD36 promoter region. The results from CHIP assay showed that histone could bind the predicted five CD36 promoter targets and that RXRA had the highest enrichment, at −860/−852, −688/−680, and −165/−157 sites, compared with the negative control IgG group (Figure 3F). RXRA could promote the expression of CD36 by binding to these sites.

### 2.5. The CD36 Inhibits the Accumulation of TGs, CHOL, and NEFAs in Myoblasts

To explore the effect of CD36 on lipid content, the CD36 of CS2 cells was overexpressed and interfered in. The transfection result can be found in Appendix A. The results showed that compared with cells transfected with the empty vector, CD36 expression was effectively overexpressed and knocked down after oeCD36 and shCD36 vector transfection (Figure 4A). Then, western blot analysis was performed to verify the level of CD36 protein expression 24 h after the transfection of oeCD36 and shCD36. CD36 overexpression was confirmed in CS2 cells transfected with oeCD36 by western blot analysis (Figure 4B,C). In addition, CD36 expression was effectively knocked down after shCD36 vector transfection (Figure 4B,C).

To explore the role of CD36 in lipid metabolism of CS2 cells, the TG, CHOL, and NEFA contents in the CS2 cells were measured. Compared with the overexpression and interference empty vector control groups, the TG content in the oeCD36 group was reduced, and the TG content in the shCD36 group was increased (Figure 4D). The CHOL content in the oeCD36 group was lower than that in the oeNC group, while the CHOL content in the shRXRA group was increased (Figure 4E). The NEFA content in the oeCD36 group was lower than that in the oeNC group; in contrast, the NEFA content in the shCD36 group was higher than that in the shNC group (Figure 4F). The number of fat droplets in the oeCD36 group was lower than that in the oeNC group. However, the number of fat droplets increased after the knockdown of CD36 (Figure 4G,H). CD36 knockdown in RXRA overexpressing cells and CD36 overexpression in RXRA knockdown cells showed no change in TGs compared with controls (Figure 4I). These results indicated that RXRA inhibits lipid accumulation in CS2 cells by promoting the expression of CD36.

## 3. Discussion

Compared with other tissues, the skeletal muscle tissue of meat ducks consumes the most energy and is the center of fat metabolism [18]. RXRA is one of the most widely distributed transcriptional regulators, and it participates in the expression of key genes in lipid metabolism [19]. Different RXR-selective agonists affected TGs in rat serum [20,21,22]. The degree to which the agonists of RXRA cause acute hypertriglyceridemia differs by the duration of agonist exposure [23]. The accumulation of TGs in skeletal muscle is closely associated with insulin resistance and type 2 diabetes [24]. The RXRA gene is related to the fat metabolism process through PPARG induction, such as TG formation and fatty acid oxidation [25]. The RXRA promoted the FA β oxidation of PPARG and reduced the accumulation of TGs in cells by increasing PPARG expression. PPARG can induce the expression of ACSL 1, which plays an important role in lipids breakdown [26]. Stossi double-labeled RXRA and lipid in human liver cells and found that the RXRA level was negatively correlated with lipid content [11]. The ELOVL6 gene plays an important role in the synthesis of long-chain saturated and monounsaturated fatty acids [27,28]. In this study, RXRA promoted the expression of ELOVL6, but the pathway regulating fatty acid synthesis is still unclear. In addition, RXRA has been previously shown to play a key role in CHOL homeostasis, intestinal CHOL absorption, and bile acid synthesis [21]. Rexone (LG268) activates RXRs to form specific dimers, which inhibit cholesterol absorption and reverse cholesterol transport through the expression of the antiporter ABC1 and the bile synthesis rate-limiting enzyme CYP7A1 [29]. ACBA1 is a member of the ABC transporter family, and this member transfers CHOL from the periphery to ester-free apolipoprotein A1 to form high-density lipoprotein (HDL) [30]. The overexpression of ABCA1 in mice increased plasma HDL levels and prevented atherosclerosis [31]. Notably, ABC1 is a target gene of RXRA, and an analysis of an ABC1-knockout mouse model produced results similar to those of the RXR-knockout model [32,33]. Therefore, RXRA is closely related to lipid metabolites such as TGs, CHOL, and NEFAs in CS2 cells.

Many transcriptions factor and transporters are involved in lipid metabolism. These transcription factors and transporters are regulated by the signal transduction pathway, forming a complex and fine-tuned regulatory network that maintains the lipid metabolism balance in cells and the whole organism. ACSL1 is induced by PPARs and plays an important role in the fatty acid β oxidation process in skeletal muscle [34]. The ELOVL6 gene in the PPAR-signaling pathway also participates in lipid metabolism [35]. RXRA is involved in lipid metabolism in the PPAR-signaling pathway. The target relationship between RXRA and FABPA genes has been confirmed during adipocyte differentiation [36]. CD36 plays an important role in regulating lipid and energy metabolism downstream of the PPAR-signaling pathway [37]. The ligand activation of the RXRA heterodimer in myelomonocytic cell lines promoted the uptake of oxLDL through the transcriptional induction of the CD36 [38]. Furthermore, RXR agonists, which upregulate CD36 expression through direct promoter interactions, were shown to induce an increase in CD36-mediated phagocytosis and a decrease in malaria-induced TNF-α secretion by human monocytes and macrophages [39,40]. Such evidence indicates that the RXRA-CD36-mediated regulation of lipid is not conservative in other cell lines. Research on the relationship between RXRA and CD36 in duck myoblasts has not been published. Therefore, the PROMO software was used for the RXRA transcription factor binding sites in the CD36 promoter region. In this study, different experiments have proved that RXRA combines with the CD36 promoter region to promote the expression of CD36.

CD36 is a single-chain transmembrane surface glycoprotein belonging to the Class B scavenger receptor family [41]. CD36 is a multifunctional membrane protein that facilitates the uptake of long-chain fatty acids [26]. Accumulating evidence suggests that CD36 is involved in the regulation of intracellular signal transduction that modulates fatty acid usage [42]. CD36 can activate fat degradation, and ACADVL, ACADM, and HADHA participate in this process [43]. In addition, CD36 recognizes many lipid ligands and combines with oxidized lipoproteins to coordinate lipid metabolism [44]. CD36 has also been proved to contribute to carbohydrate and lipid metabolism, together with fatty acids [45,46]. Previously, Ibrahimi established a model of CD36 overexpression by activating the creatine kinase (MCK) promoter in mouse muscle, and the results showed that the transgenic mice lost weight and adipose tissue and exhibited significantly reduced TGs, CHOL, and FAs and very-low-density lipoprotein (VLDL) levels [47]. Similarly, CD36 plays an important role in liver lipid homeostasis. The function of the CD36 gene in duck myoblasts still needs further exploration.

On the basis of the results of this study, a model showing the inhibition of lipid accumulation in CS2 cells as mediated by RXRA and CD36 was proposed. In this model, RXRA can combine with the CD36 promoter region to promote the expression of CD36 and inhibit lipid accumulation in CS2 cells. Although the fact that RXRA and CD36 comprise a regulatory network was established, other components in CS2 cells may be involved in this network. This provides a platform for further understanding the lipid metabolism of duck myoblasts.

## 4. Materials and Methods

### 4.1. Animals

In order to study the expression of RXRA in different tissues, the breast muscles and other tissues of 42-day-old male Qiangying ducks were collected. The collected tissues were stored in an ultralow-temperature refrigerator at −80 °C for RNA extraction.

### 4.2. RNA Isolation and Quantitative Real-Time PCR (RT-PCR)

Total RNA was extracted from duck heart, liver, spleen, lung, kidney, subcutaneous fat, abdominal fat, breast muscle, leg muscle tissue, and cells with TRIzol (Invitrogen, Carlsbad, CA, USA) following the supplier’s protocol. Here, 1 μg of total RNA was extracted with an RNA extraction kit. Further, cDNA was reverse transcribed using the RT-cDNA synthesis kit. The reaction system was as follows: 95 °C for 5 min, 95 °C for 30 s, and 60 °C for 30 s—35 cycles. RT-PCR (ABI7500, Waltham, MA, USA) was performed with qPCR mix. With GAPDH as the reference gene, RT-PCR was used to detect the mRNA levels. The 2-ΔΔct method was used to calculate the relative expression of the genes expressed in cells. According to the KEGG database, RXRA regulated the lipid metabolism by participating in the PPAR-signaling pathway (map03320). The involved lipid metabolism-related gene primers (Appendix A) were designed according to the PPAR-signaling pathway.

### 4.3. Primary Myoblasts Separation, Culture, and Differentiation

The separation and culture methods used to obtain myoblasts (CS2 cells) were described in a previous study [48]. Dulbecco’s modified Eagle’s medium/nutrient mixture F-12 (DMEM/F12, VivaCell, Shanghai, China), containing 10% fetal bovine serum (FBS, VivaCell) and a 100 U/mL penicillin–streptomycin solution (VivaCell), was used to culture the CS2 cells.

CS2 cell differentiation was induced by growth medium exchange for the differentiation medium (DMEM/F12 with 2% horse serum and 1% of penicillin/streptomycin). The differentiation media was changed daily, and the samples were collected after 0, 2, 4, and 6 days.

### 4.4. Immunofluorescence

When cells grew to 80% confluence, they were reseeded into 24 well plates for immunofluorescence staining. The cells were fixed for 15 min with 4% paraformaldehyde (Biosharp, Hefei, China), blocked, and incubated with antipaired box 7 (PAX7, Proteintech, Wuhan, China), MYOD (Proteintech, Wuhan, China), and RXRA (Proteintech, Wuhan, China) primary antibody at 4 °C overnight. The primary antibody was detected with a secondary antibody conjugated to fluorescein isothiocyanate (FITC) (Immunoway, Suzhou, China) in a dark room at 37 °C for 1 h. Finally, the cells were incubated with DAPI (Servicebio, Wuhan, China) in the dark for 10 min. The immunofluorescence images were then visualized with a confocal microscope (DP73, Olympus, Tokyo, Japan). ImageJ was used to count the positive cells.

### 4.5. Western Blotting

Cells transfected with plasmids were harvested 24 h after transfection. The cells were then resuspended in RIPA buffer with protease inhibitors (Meilunbio, Dalian, China). The cell lysate was removed by centrifugation at 12,000 rpm/min for 10 min at 4 °C, and the supernatant was collected. The BCA (Vazyme, Nanjing, China) protein quantitative analysis method was used to determine the total protein concentration in the samples. For western blotting (Bio-Rad, Berkeley, CA, USA), the same amount of protein was separated by SDS–PAGE and then transferred to PVDF (Thermo, Waltham, MA, USA) membranes with a gel-to-membrane transfer system. After blocking with 1× PBS buffer (VivaCell, Shanghai, China) containing 1% bovine serum albumin (BSA, Servicebio, Wuhan, China) for 2 h, the PVDF membrane was washed with a TBST (Sangon Biotech, Shanghai, China) solution. Then, the membrane was incubated with primary antibody (diluted in 1% BSA) at 4 °C for 12 h. Primary antibodies against RXRA (YN0018), GAPDH (YM3215), FABP4 (YM0013), ELOVL6 (YT1540), and ACSL1 (YN0827) were all purchased from ImmunoWay, China; PPARG (16643-1-AP) was purchased from Proteintech, China; and CD36 (bs-8873R) was purchased from Bioss. The membrane was incubated with a secondary antibody (ImmunoWay) (1:1000 diluted in 1% BSA) for 2 h. The signal was visualized by using an enhanced chemiluminescence kit (Vazyme). The relative intensities of the bands were quantified with ImageJ software (Bethesda, Rockville, MD, USA).

### 4.6. Analyses of the Intracellular Triglyceride (TG), Cholesterol (CHOL), and Nonesterified Fatty Acid (NEFA) Contents

Following the respective manufacturer’s instructions, we used kits to determine the TG (Applygen, E1013-105), CHOL (Applygen, E1015-105), and NEFA (Nanjing Jiancheng, Nanjing, China, A042-2-1) contents. The CS2 cells were collected 24 h after transfection with the overexpression plasmid or the interference plasmid. Different samples were adjusted to the same cell level through BCA protein quantification, and the test for each model was repeated three times. The TG, CHOL, and NEFA levels in the cells were adjusted according to the protein content. A microplate reader was used to determine the concentrations of TGs, CHOL, and NEFAs.

### 4.7. Vector Construction

RNA from CS2 cells were extracted by using the TRIzol (Invitrogen, 15596018) method. RNA was reverse transcribed into cDNA according to the instructions in a first-strand synthesis premix kit (Vazyme, R122-01). Full-length RXRA (XM_027471073.2) was amplified with specific primers (Appendix A), including the NheI (NEB, R3131S) and HindIII (NEB, R3104 V) restriction sites. The pBI-CMV3 vector and an RXRA fragment were digested with NheI and HindIII. OeRXRA was constructed by linking the purified pBI-CMV3 vector with the RXRA fragment. OeRXRA was transfected into CS2 cells by using ExFect transfection reagent (Vazyme, T101-01). The transfection conditions were ExFect: oeRXRA = 2:1. Four interference targets were identified on the basis of the full-length RXRA sequence, and upper and lower primers were designed, including the AgeI (NEB, R3552S) and EcoRi (NEB, R310T) restriction sites at the 5 ft and 3 ft ends, respectively. The primers (Appendix A) were annealed to form double-stranded oligomers and were attached to the PTSB-SH-copGFP-2A-PURO skeleton to construct short interfering RNAs: shRXRA1, shRXRA2, shRXRA3, and shRXRA4. These four interference vectors against RXRA were transfected into CS2 cells. After 24 h, the cell samples were collected, and the RNA was extracted and reverse transcribed into cDNA. The effects of each interference vector on RXRA expression were quantitatively determined by RT-PCR (Vazyme, Q711-02). The shRXRA-4 and shCD36-1 with the highest interference efficiency were named shRXRA and shCD36 for all subsequent experiments (Appendix A). The above methods were also used to construct CD36 (XM_038183702.1) overexpression and interference vectors. The primer synthesis in this study was completed by Shanghai Sangon Biotechnology Co., Ltd. A wild-type CD36 promoter luciferase reporter vector (wt-CD36) and binding site mutant vector (mu-CD36) were synthesized by Shanghai Gene Pharma Co., Ltd. (Shanghai, China), and cloned into a GPL3-Basic vector to form wt-CD36 and mu-CD36. A PGL3-RL vector was purchased from Shanghai Gene Pharma Co., Ltd.

### 4.8. Oil Red O Staining

Oil Red O stock solution and working solution: Oil Red O powder (0.2 g, Sigma, MA, USA) was dissolved in 40 mL of isopropanol overnight. Before staining, the stock solution was filtered with a number 1 filter. A working solution was prepared by diluting the stock solution with water with a 6:4bratio, allowing it to stand for 10 min at room temperature and then filtering it (0.22 µm, Millipore, MA, USA).

After differentiation and transfection, the CS2 cells were washed three times with PBS. Cells were fixed with 4% paraformaldehyde for 30 min and washed three times with PBS. The cells were stained with Oil Red O working solution for 30 min, the stain was discarded, the cells were washed three times with 60% isopropanol, and then the cells were washed three times with PBS. Images were taken and observed under a microscope. Then, 200 µL of 60% isopropanol was added to the sample for 20 min, and a microplate reader was used to measure the absorbance at 510 nm.

### 4.9. Luciferase Reporter Assay

The mature 2000 bp sequence of the promoter before the CD36 translation initiation site was selected for transcription factor prediction by using PROMO (http://alggen.lsi.upc.es/cgi-bin/promo_v3/promo/promoinit.cgi?dirDB=TF_8.3, accessed on 25 March 2021) software. PROMO predicted five targeted binding sites for RXRA and the CD36 promoter region. The complete CD36 promoter sequence (wt-CD36) and the CD36 promoter sequence with five target sequences missing (mu-CD36) were submitted to Gemma biosynthesis, which ligated them into a GPL3-Basic cloning vector. OeRXRA/oeNC/wt-CD36/mu-CD36/GPL3-Basic was cotransfected into CS2 cells, and a Renilla luciferase vector (PGL-RL) was used to correct the transfection efficiency data. The firefly (hluc+) and Renilla (hRluc) luciferase intensities were detected with a Dual-Glo luciferase assay system (Promega, WI, USA, E1960) according to the manufacturer’s protocols, and hluc+ luciferase was used as the reference to correct for variations in transfection efficiency. The relative luciferase activity was calculated with the following equation: relative luciferase activity = hRluc/hluc+. A Bio-Tex microplate reader was used to detect the fluorescence values and determine whether the promoter of the CD36 gene contained binding sites for RXRA.

### 4.10. Chromatin Immunoprecipitation (CHIP)

CHIP-IT High Sensitivity kit (Active Motif, Carlsbad, CA, USA) was used to determine the association of transcription factor RXRA with its specific genomic regions on the CD36 gene. CS2 cells were transfected with oeRXRA for 24 h and subjected to cell fixation, chromatin sonication, immunoprecipitation, and DNA purification. Anti-RXRA (21218-1-AP, Proteintech, Wuhan, China) was used for RXRA protein detection. Histone-H3 (17168-1-AP, Proteintech, Wuhan, China) was used for positive control. Rabbit IgG (300000-0-AP, Proteintech, Wuhan, China) was used as a negative control in the immunoprecipitation experiments. The immunoprecipitated fraction was analyzed by PCR and qRT-PCR to determine the abundance of the target DNA sequence(s) relative to the input chromatin.

### 4.11. Statistical Analysis

The expression of RXRA in tissues and CS2 cell at differentiation stages and the lipid content in CS2 cells at their differentiation stages were analyzed by one-way ANOVA. The expression of RXRA and CD36, genes related to lipid metabolism, and the lipid content in CS2 cells transfected with overexpressed or knockdown vectors were compared with normal controls by using a *t*-test. Firefly luciferase activity and Renilla luciferase activity were compared in each of the two groups by using a *t*-test. The efficiency of transcription factor RXRA binding to the CD36 promoter were compared with the positive control and the negative control, separately, by using *t*-tests. Data were presented as the mean ± SD. * indicates *p* < 0.05, ** indicates *p* < 0.01, *** indicates *p* < 0.001, and **** indicates *p* < 0.0001.

## 5. Conclusions

In summary, RXRA reduced the accumulation of TGs, CHOL, NEFAs, and fat droplets in duck myoblasts by promoting the expression of CD36.

## Figures and Tables

**Figure 1 ijms-24-01180-f001:**
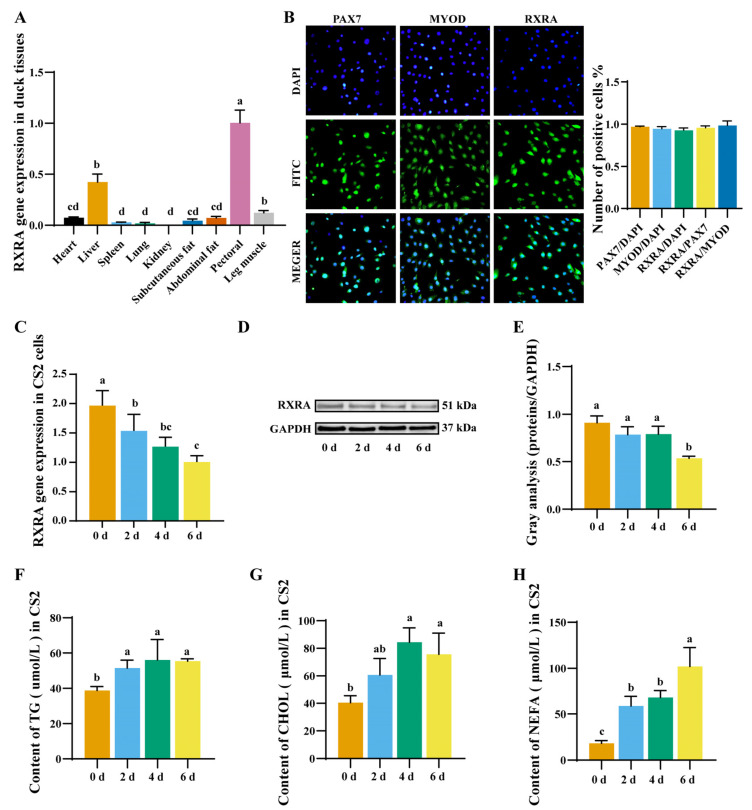
Expression level of RXRA gene in duck tissue and myoblasts differentiation. (**A**) The expression level of the RXRA gene in duck tissues. (**B**) Immunostaining with anti-PAX7, anti-MYOD, and anti-RXRA antibodies, and the ratio of positive cells. The images were taken under 20×. (**C**) The expression of RXRA gene mRNA in myoblasts at 0 days, 2 days, 4 days, and 6 days of differentiation. (**D**) The expression of RXRA protein in myoblasts at 0 days, 2 days, 4 days, and 6 days of differentiation. (**E**) The relative fold change in RXRA/GAPDH as determined by western blotting was quantified through a gray scale scan. The contents of TGs (**F**), CHOL (**G**), and NEFAs (**H**) in myoblasts at 0 days, 2 days, 4 days, and 6 days of differentiation. ^a–d^ different superscripts mean a significant difference (*p* < 0.05).

**Figure 2 ijms-24-01180-f002:**
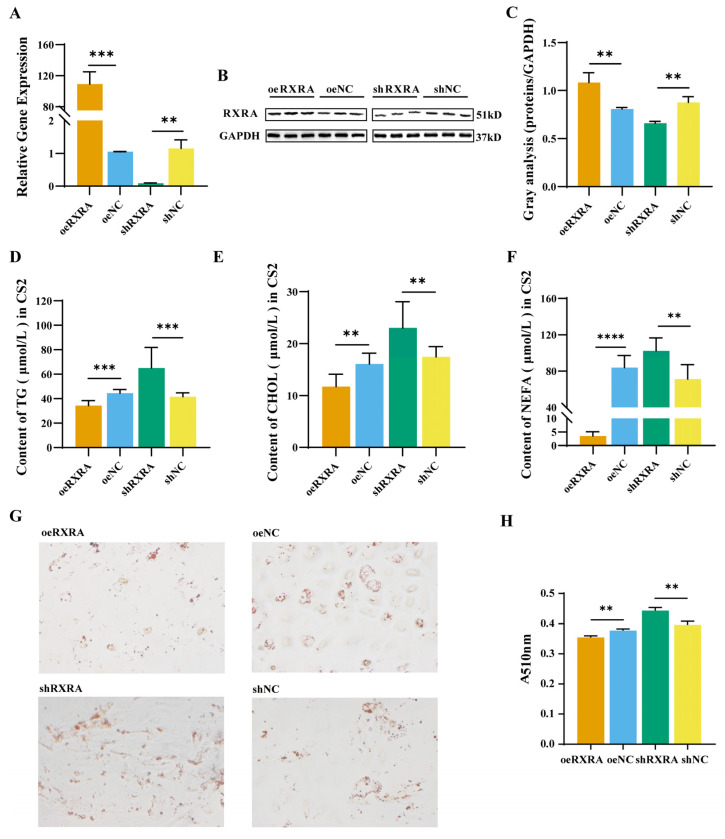
RXRA gene inhibits the contents of TGs, CHOL, and NEFAs in myoblasts. (**A**) Relative mRNA expression levels of RXRA in CS2 cells transfected with oeRXRA, oeNC, shRXRA, and shNC. (**B**) The indicated protein levels were detected by western blot analysis. (**C**) The relative fold change in RXRA/GAPDH as determined by western blotting was quantified through a gray scale scan. TG (**D**), CHOL (**E**), and NEFA (**F**) contents in CS2 cells transfected with overexpression vectors oeRXRA and oeNC and short hairpin (interference) vectors shRXRA and shNC. (**G**) Lipid droplets in CS2 cells. The images were taken under 20×. (**H**) Quantitative results of lipid droplets in CS2 cells. The data were reported as the means ± SD on the basis of three experiments. ** indicates *p* < 0.01, *** indicates *p* < 0.001, and **** indicates *p* < 0.0001.

**Figure 3 ijms-24-01180-f003:**
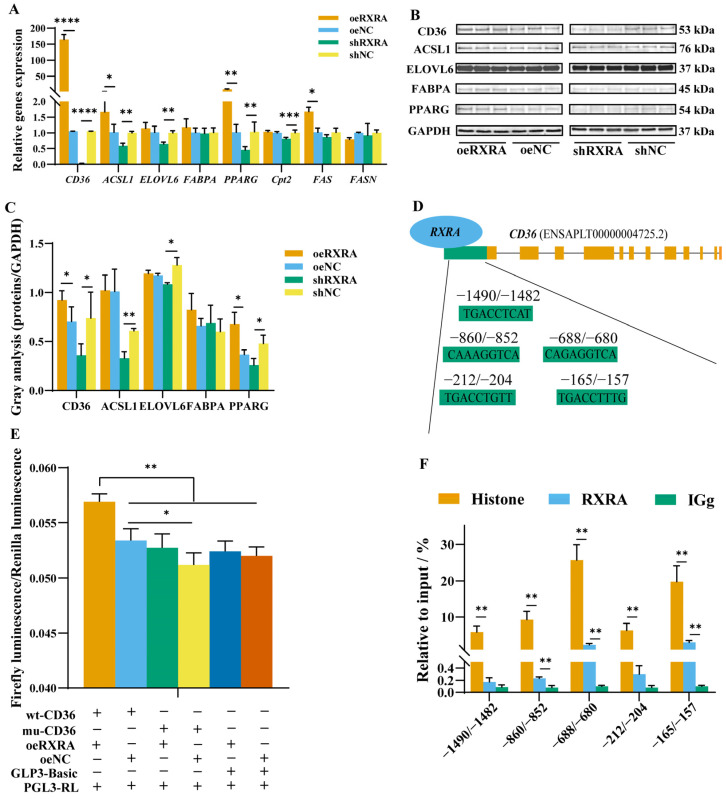
Effect of RXRA on lipid metabolism-related gene and protein expression in myoblasts. (**A**) The gene expression levels of CD36, ACSL1, ELOVL6, FABPA, PPARG, Cpt2, FAS, and FASN after the transfection of overexpression vectors oeRXRA and oeNC and short hairpin (interference) vectors shRXRA and shNC in CS2 cells. (**B**) The indicated protein levels were detected by western blotting. (**C**) The relative changes in CD36, ACSL1, ELOVL6, FABPA, and PPARG expression (protein/GAPDH) in the western blots were quantified by the gray scale scan and reported as the fold change. (**D**) Prediction of RXRA-binding sites in the CD36 promoter region. (**E**) Analysis of firefly luciferase activity and Renilla luciferase activity. (**F**) CHIP analysis of transcription factor RXRA binding to CD36 promoter target sites. The data are reported as the mean ± SD on the basis of three experiments. * indicates *p* < 0.05, ** indicates *p* < 0.01, *** indicates *p* < 0.001, and **** indicates *p* < 0.0001.

**Figure 4 ijms-24-01180-f004:**
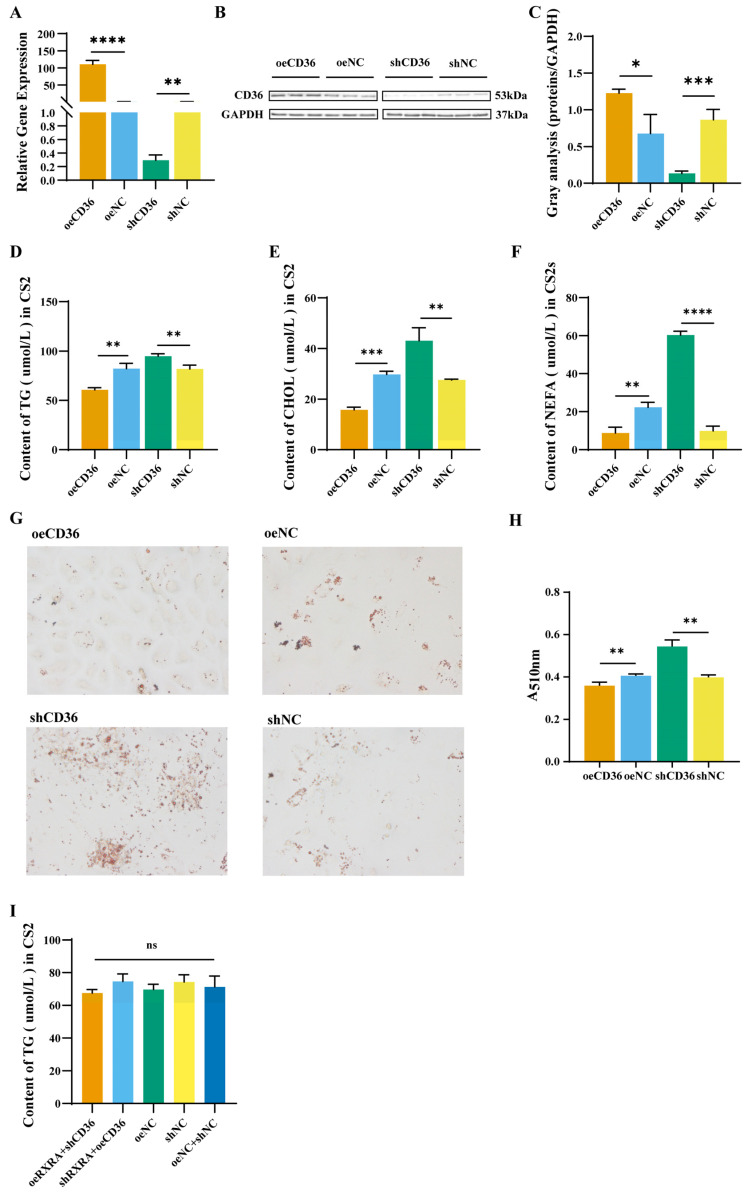
CD36 gene inhibits the contents of TGs, CHOL, and NEFAs in myoblasts. (**A**) Relative mRNA expression levels of CD36 in CS2 cells transfected with oeCD36, oeNC, shCD36, and shNC. (**B**) The indicated protein levels were detected by western blot analysis. (**C**) The relative fold change in CD36/GAPDH as determined by western blotting was quantified through a gray scale scan. TG (**D**), CHOL (**E**), and NEFA (**F**) contents in CS2 cells transfected with overexpression vectors oeRXRA and oeNC and short hairpin (interference) vectors shRXRA and shNC. (**G**) Lipid droplets in CS2 cells. The images were taken under 20×. (**H**) Quantitative results of lipid droplets in CS2 cells. (**I**) TG contents of CD36 knockdown in RXRA overexpression cells and CD36 overexpression in RXRA knockdown cells. The data were reported as the means ± SD on the basis of three experiments. ^ns^ indicates *p* > 0.05, * indicates *p* < 0.05, ** indicates *p* < 0.01, *** indicates *p* < 0.001, and **** indicates *p* < 0.0001.

## Data Availability

The data related to this paper may be requested from the corresponding author. More data supporting reported results can be found in Appendix A.

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
