# Peer review of "A Novel in Duck Myoblasts: The Transcription Factor Retinoid X Receptor Alpha (RXRA) Inhibits Lipid Accumulation by Promoting CD36 Expression"

_ijms, 2023, doi:10.3390/ijms24021180_

Round 1
Reviewer 1 Report
In this work Pan et al describe the role of RXRA in the regulation of lipid metabolism in primary duck myoblasts. They use a combination of over-expression and knockdown experiments to investigate the contribution of this key nuclear receptor in inhibiting lipid accumulation. In addition they identified the transmembrane protein CD36 as a downstream target of RXRA in duck myoblasts by luciferase, ChIP and gene expression analysis. Finally, the authors also addressed the role of CD36 in muscle cells through over-expression and knockdown experiments.
The work is scientifically sound and well-performed and significantly increases our knowledge on the role of the RXR-CD36 axis in regulating lipid metabolism in muscle cells. However, some minor points should be addressed before publication in IJM
1) The immunofluorescences in figure 1 are barely visible. The authors should provide higher magnification images and the number of positive cells should be quantified. Are all Pax7-positive cells also RXR-positive ? Is RXRA signal nuclear in all cells?
2) It is not clear what a,b,c means in the statistical analysis in figure 1. The authors should explain it better. Also in all the other figures they used asterisks for the statistical analysis. Why is figure 1 different?
3) In the ChIP analysis (figure 3f) the authors should specify which antibodies they used for histones and RXRA, and specify which histones did they analyze. Is the histone antibody used just as an internal control for the ChIP experiments? Please specify.
4) The authors need to carefully check all the references, as some of them are misplaced (ie. ref 10)
Author Response
Thank you very much for your comments dated on November 22, 2022. For the comments of Reviewer 1, we explained and significantly revised the paper.
Please see the attachment.

Reviewer 2 Report
In this article, Pan et al report that RXRa acts by regulating CD36 expression in duck myoblasts to maintain lipid homeostasis. Although the role of RXRa is regulating lipid metabolism has been shown in mice and other cell lines, it is novel to study it’s role in duck myoblasts. These are the following comments that need to be addressed.
Major Comments:
1. 1. The protein knockdown as well as overexpression of RXRa is not robust. Isolate nuclei from these cells and then perform western blotting.
2. 2. Check gene expression of other fat metabolism genes like Cpt1/2, and check if there is a change in fatty acid synthesis pathway.
3. 3. The authors show that RXRa and CD36 overexpression individually lowers the triglyceride levels. To validate that it is indeed the pathway, the authors must perform CD36 overexpression in RXR knockdown cells and vice versa.
4. 4. Is the RXRa - CD36 mediated regulation of lipid homeostasis conserved in myoblasts across the species? Show this pathway in at least a human cell line.
Minor Comment:
1. 1. There are quite a few typos that need to be fixed. Please proof-read.
Author Response
Thank you very much for your comments dated on November 22, 2022. For the comments of Reviewer 2, we explained and significantly revised the paper.
Please see the attachment.

Round 2
Reviewer 1 Report
I appreciate the authors’ efforts to address my suggestions. However, in the light of their answers some concerns have arisen.
Point 1, the higher magnifications of Figure 1b have now revealed that the immunofluorescence signal for Pax7 is not nuclear. This is very strange as this transcription factor is exclusively found in the nucleus of muscle cells. It seems to me that the staining presented is only background signal, which antibody was used for these experiments? To claim the isolated cells are indeed myoblasts, stronger experimental evidence needs to be provided, either by improving Pax7 staining, by using a different antibody (ie. MyoD) or by showing their differentiation potential.
Point 2, it is not clear to me what a, b, c means in the statistical analysis in figure 1. The answer “Significant differences (p <0.05) are shown in superscript letters, with different letters representing significant differences” does not explain what a, b or c refer to. Please specify.
Point 3, the authors described which H3 antibody they used but they also included the following sentence: “ his-tone cluster 2 (H3a) was analyzed”. What does this refer to? There is no indication to this cluster in the figure.
In general, more details on the experimental methodology and analysis should be provided, including all antibodies, experimental protocols and analysis used.
Author Response
Thank you very much for your comments dated on 01 December 2022. For the comments of Reviewer 1, we explained and significantly revised the paper.
Please see the attachment.

Reviewer 2 Report
Thank you for your response to my comments.
Author Response
Thank you for your review and hard work.
Round 3
Reviewer 1 Report
The authors have now addressed my technical questions. However, their answer to my point 2 is still unaddressed and it is really not clear what a, b or c mean in the figure. Their answer: "Different lowercase letters (a, b or c) in the same figure indicate that the data are different (p < 0.05)" is indeed very vague. Could you please indicate what do a, b and c refer to precisely? Are they all the same thing? And if so why to use different letters rather than asterisks?
Author Response
Thank you very much for your comments dated on 06 December 2022. For the comments of Reviewer 1, we explained and significantly revised the paper.
Please see the attachment.

Round 4
Reviewer 1 Report
Thanks for the explanation